# Attitudes of Patients with Adrenoleukodystrophy towards Sex-Specific Newborn Screening

**DOI:** 10.3390/ijns9030051

**Published:** 2023-09-02

**Authors:** Hemmo A. F. Yska, Lidewij Henneman, Rinse W. Barendsen, Marc Engelen, Stephan Kemp

**Affiliations:** 1Department of Child Neurology, Amsterdam Leukodystrophy Center, Emma Children’s Hospital, Amsterdam UMC Location University of Amsterdam, Amsterdam Neuroscience, 1105 AZ Amsterdam, The Netherlands; h.a.f.yska@amsterdamumc.nl (H.A.F.Y.); m.engelen@amsterdamumc.nl (M.E.); 2Department of Human Genetics, Amsterdam Reproduction and Development Research Institute, Amsterdam UMC Location Vrije Universiteit Amsterdam, 1081 HV Amsterdam, The Netherlands; l.henneman@amsterdamumc.nl; 3Laboratory Genetic Metabolic Diseases, Department of Clinical Chemistry, Amsterdam Gastroenterology Endocrinology Metabolism, Amsterdam UMC Location University of Amsterdam, 1105 AZ Amsterdam, The Netherlands; r.w.barendsen@gmail.com

**Keywords:** adrenoleukodystrophy, newborn screening, sex-specific screening, survey, Wilson and Jungner

## Abstract

Newborn screening (NBS) for X-linked adrenoleukodystrophy (ALD) can identify affected individuals before the onset of life-threatening manifestations. Some countries have decided to only screen boys (sex-specific screening). This study investigates the attitudes of individuals with ALD towards sex-specific NBS for ALD. A questionnaire was sent to all patients in the Dutch ALD cohort. Invitees were asked who they thought should be screened for ALD: only boys, both boys and girls or neither. The motives and background characteristics of respondents were compared between screening preferences. Out of 108 invitees, 66 participants (61%), 38 men and 28 women, participated in this study. The majority (n = 53, 80%) favored screening both newborn boys and girls for ALD, while 20% preferred boys only. None of the respondents felt that newborns should not be screened for ALD. There were no differences in the background characteristics of the respondents between screening preferences. Our study revealed a diverse range of motivations underlying respondents’ screening preferences. This study is one of the first to investigate the attitudes of patients towards sex-specific screening for ALD. The outcomes of this study can offer insights to stakeholders engaged in the implementation of NBS programs. ALD patients are important stakeholders who can provide valuable input in this process.

## 1. Introduction

X-linked adrenoleukodystrophy (ALD) (OMIM # 300100) is a genetic disorder caused by pathogenic variants in the *ABCD1* gene, which affects the metabolism of very-long-chain fatty acids [1]. In boys, ALD can present as adrenal insufficiency or as a leukodystrophy (cerebral ALD). Early diagnosis in a presymptomatic stage allows for initiation of potential lifesaving treatments, such as steroid replacement therapy and hematopoietic stem cell transplantation (HSCT) [2]. Males with ALD are, therefore, monitored on a regular basis using endocrinological tests and MRI. Females generally do not develop life-threatening complications [3]. In adulthood, ALD causes a slowly progressive myelopathy with a highly variable age of onset in both males and females [2]. For myelopathy, there is currently no disease-modifying treatment.

Newborn screening (NBS) for ALD can identify affected individuals before the onset of life-threatening manifestations and has been incorporated into the screening programs of Taiwan and >35 states in the USA [4,5,6,7,8]. After conducting a regional pilot study in the Netherlands to evaluate the feasibility and effectivity of a diagnostic algorithm for ALD in the Dutch newborn population, nationwide screening will start in October 2023 [9,10]. Italy and Japan have taken the first exploratory steps to do the same [11,12], but the difference in disease manifestations between males and females complicates the introduction of ALD in NBS programs. The reason for this is that, according to the internationally acknowledged Wilson and Jungner criteria, screening for a disease should only be performed if there is an important health problem and an accepted treatment is available [13]. As women with ALD generally only develop the non-treatable myelopathy in adulthood, they would not be considered eligible for screening. For this reason, the Health Council of the Netherlands recommended that only boys should be screened for ALD [14]. This principle is referred to as “sex-specific screening”. Although females cannot be treated, several indirect advantages of screening girls to establish an early diagnosis of ALD can be considered from a patient perspective (Table 1) [15]. Balancing between the “ALD experiential perspective”, which is shaped by personal experiences, participation in patient organizations or by the experiences of relatives, and the public health/medico-ethical perspective guided by the Wilson and Jungner criteria, poses a challenge. From a screening perspective, sex-specific screening can be regarded as the best option, while in the eyes of individual parents and families, who have experienced the disease themselves, screening girls can be considered beneficial.

Parallel to the start of the Dutch ALD pilot study, attitudes towards the expansion of NBS with additional disorders, including ALD, were assessed among health professionals and parents of newborns [16,17,18]. A group whose views on this topic have only been assessed to a limited extent, but that may provide crucial insights on sex-specific newborn screening for ALD, is affected individuals themselves [19,20]. Men and women with ALD understand the burden of knowing early about this potentially life-threatening illness and can relate it to their own experiences. We, therefore, assessed their attitudes towards the addition of ALD to NBS. We also investigated underlying motives and the influence of background characteristics on screening preference. The results of this study can inform policy decisions and are of importance for all countries and states considering the addition of ALD, and other X-linked conditions, to their NBS programs.

## 2. Materials and Methods

### 2.1. Participants and Procedure

This study was approved by the local institutional ethics review board (W22_430 # 22.508). A cross-sectional survey study using an online questionnaire was performed on all adult (age > 18 years) patients in the Dutch ALD cohort in 2023. This cohort consists of 62 men and 46 women who are actively followed on a regular basis at the Amsterdam University Medical Centers, University of Amsterdam. Participants received an e-mail explaining the purpose of this study and provided informed consent prior to participation. A reminder was sent after two weeks in case the questionnaire was not completed after the first invitation.

### 2.2. Questionnaire Design

A questionnaire was constructed by a multi-disciplinary team of experts in the field of clinical care for ALD, newborn screening and survey studies (Appendix A). The questionnaire was tested on one person with ALD and two controls to identify errors. Respondents were asked who, in their opinion, should be screened for ALD: only boys, both boys and girls or neither. This question was derived from a previous study (PANDA study) that evaluated the attitudes of a general population of parents of newborns towards the addition of multiple disease scenarios to NBS, including ALD [18]. The question also included free-text space where participants were asked to explain the most important motives for their choice. The question was followed by 19 statements related to NBS for ALD. Participants were asked to indicate their level of agreement to each of these statements on a 5-point scale (“0” = fully disagree, “5” = fully agree). Most statements were derived from previous survey studies that evaluated patient perspectives on the addition of other diseases to NBS [21,22]. ALD-specific statements were added to the questionnaire.

We investigated a number of background characteristics. The general attitude of participants towards NBS was assessed using two questions derived from the PANDA study [18]. Participants were asked on a 5-point scale to indicate whether they considered the Dutch newborn screening bad (1)–good (5) and useless (1)–useful (5). Information on education level, country of origin, religion, alternative medicine and vaccination status of children was collected and recoded according to the Dutch Bureau of Statistics (CBS) [23]. Level of education was regrouped into three categories (low, middle and high education). Perceived disease severity of ALD was evaluated using the Brief Illness Perception Questionnaire (BIPQ). This validated questionnaire contains nine statements to which participants are asked to indicate their level of agreement on a 10-point scale [24]. We excluded one question from the original BIPQ (“What do you think are the most contributing factors to your disease”) as we considered it unsuitable for the present study. A total score (min 0–max 80) was calculated based on the provided answers, where higher scores indicated higher perceived disease severity.

### 2.3. Statistical Analysis

Incomplete questionnaires were only included if the question on who should be screened was answered. Open-text responses on motives for screening preference were coded and categorized by one of the researchers (HY). Background differences between screening preferences were compared using independent *t*-tests for age and perceived disease severity (BIPQ) and Chi-square tests for variables with nominal and ordinal data (sex, deceased family member because of ALD, alternative lifestyle, religion and education level). The selected background variables under analysis were considered most relevant based on results of the PANDA study [18]. The mean levels of agreement for each of the 19 statements following the question on who should be screened were compared between screening preferences in order to identify decisive motives. In order to explore differences in levels of agreement between sexes, we performed independent *t*-tests for each statement. An α-level of significance was set at *p* = 0.05 for all tests. We used IBM SPSS statistics (version 28.0) for our analyses.

## 3. Results

### 3.1. Respondent Characteristics

Out of 108 invitees, 67 completed the questionnaire. One respondent was excluded from analysis since the question on screening preference was left unanswered. This resulted in 66 of 108 (61%) included participants. A total of 38 of 62 men (61%) and 28 of 46 women (61%) participated. Table 2 describes the characteristics of the respondents. Females were older than males (54 ± 14 years vs. 46 ± 17 years, respectively). Most patients were diagnosed with ALD at an adult age (65%) and most often after another family member tested positive (64%). The initial diagnosis in females was only sporadically (7%) established after ALD-related symptoms as opposed to 47% of males. The majority of respondents (52%) reported the death of a family member due to ALD. Almost half of the respondents had a high level of education (48%).

### 3.2. Attitude towards the Addition of ALD to NBS

The majority of respondents (n = 53, 80%) were in favor of screening both newborn boys and girls for ALD (Table 3). Thirteen respondents (20%) were in favor of sex-specific screening (i.e., boys only). There was no difference in screening preference between men and women. None of the participants indicated that they were against NBS for ALD.

### 3.3. Open-Text Responses

In the open-text fields, the most important reasons reported by participants who felt that only boys should be screened were that boys develop more severe symptoms (6/13, 46%) and that girls develop no or less severe symptoms (3/13, 23%) (Appendix A). One respondent stated: “Boys experience the most problems that are hard to live with”. Important reasons given by respondents who felt that both boys and girls should be screened were that it helps to provide an early diagnosis (13/48, 27%), it facilitates (early) treatment and monitoring (12/48, 25%) and that it could help with family planning (8/48, 17%). One respondent stated: “The consequences of [a girl] not knowing you have ALD can be very severe. With newborn screening and its results, you can get better support and disease monitoring”. Other reasons given were that, according to respondents, the disease has an impact on both sexes (7/48, 15%), the possibilities knowing about ALD provide for future planning (6/48, 13%) and that a test should not discriminate between sexes (3/48, 6%).

### 3.4. Agreement with Statements

All respondents agreed with statements in favor of screening boys (mean 4.08–4.54) and disagreed with statements opposed to screening newborns in general (mean 1.38–2.77) (Table 4). Respondents in favor of only screening boys gave positive scores (mean 3.54–3.77) to arguments opposed to screening newborn girls. These respondents agreed most with the statement “ALD should not be detected early in girls because there is no treatment available for their symptoms” (mean 3.77). They were slightly positive about the early detection of ALD in girls “because it enables the identification of other family members with ALD (including boys and men)” (mean 3.38) and because “parents can be informed in time about possibilities for further family planning” (mean 3.46).

Respondents in favor of screening both boys and girls gave positive scores to all arguments in favor of screening girls (mean 3.78–4.25) and negative scores to arguments opposed to screening girls (mean 1.71–1.94). The statement “ALD should be detected early in girls because it enables the identification of other family members with ALD (including boys and men)” received the highest support (mean 4.25) from this group. The statement “ALD should NOT be detected early in girls, because they almost never develop life-threatening symptoms” received the lowest support (mean 1.71).

Independent *t*-tests for each statement revealed no significant differences in the mean levels of agreement between male and female respondents (data not shown).

### 3.5. Differences in Background between Screening Preferences

There were no significant differences between screening preferences for age (*p* = 0.968), perceived disease severity (BIPQ) (*p* = 0.842), sex (*p* = 0.747), if a family member had died because of ALD (*p* = 0.619) or if respondents valued an alternative lifestyle (*p* = 0.118). We were unable to perform statistical tests for religion and education level since the assumptions for the Chi-square test were not met for these variables [25]. The mean scores for the questions on whether respondents considered NBS bad or good and useless or useful were 4.85 for the respondents in favor of screening both boys and girls and 4.62 for the respondents in favor of sex-specific screening.

## 4. Discussion

This study assessed the attitudes of individuals with ALD towards (sex-specific) NBS for ALD. Importantly, none of the respondents felt that ALD should be excluded from NBS, and the majority (80%) were in favor of screening both boys and girls. The background characteristics of respondents, including sex, did not differ between respondents with different screening preferences. Furthermore, we identified a diverse range of agreement levels with statements on why respondents preferred a certain type of screening.

It is not surprising that most patients will be in favor of screening for their disease, but few studies have investigated the views of ALD patients on NBS for ALD. In 1991, Costakos et al. conducted a study to evaluate the attitudes of patients and their family members and found a high level of support for NBS for ALD [19]. Their results are difficult to compare to our work because of the long period between the studies and the developments in diagnostic tools and treatments (such as HSCT) since that time. Another study from 2007 by Schaller et al. focused on the attitudes of family members of people with ALD in the United States. It found that 112/128 (88%) respondents felt that both boys and girls should be screened for ALD. Of the 16 (13%) respondents who did not support the screening of boys and girls, 10 (7.8%) respondents thought the choice should be left to the parents, 2 (1.6%) did not support screening for ALD and 1 (0.8%) supported only screening boys [20]. In our study, we found a much larger group of respondents in favor of sex-specific screening. This dissimilarity may be explained by differences in preference between patients and family members or by differences between study methods. The proportions of screening preferences in our study are similar to those recently described by van der Pal et al. (2022) in a general population of 804 parents of (healthy) newborns who participated in NBS in the Netherlands [18]. Of the large majority (796) in favor of screening for ALD, 665 (84%) considered it best to screen both sexes, and 131 (16%) thought it would be best to screen only boys.

The Wilson and Jungner criteria are considered guiding principles for determining the diseases for which newborns should be screened [13,16,26]. Failing to adhere to these criteria could lead to reduced public acceptance and participation in NBS [16,27]. In women, ALD is not life-threatening, is untreatable and a presymptomatic diagnosis has no direct health benefit. Diagnosing presymptomatic newborn girls with ALD can result in stress and uncertainty for both parents and the newborn, and it can lead to questions related to future follow-up and prognosis that may be difficult to answer [28]. Furthermore, the diagnosis of a presymptomatic genetic disorder may have financial repercussions (for instance, eligibility for life insurance). Because of the potential negative effects for females, the Health Council of the Netherlands advised against screening girls for ALD [14]. The fact that, currently, none of the >35 states in the United States that screen newborns for ALD have implemented a sex-specific screening protocol implies that ALD-specific considerations (Table 1) may outweigh the disadvantages [27,29]. Some argue that the identification of ALD in all newborns, including girls, should be preferred, as it enables extended family screening and the identification of at-risk family members [20]. Interestingly, in our study, respondents who were in favor of screening only boys also acknowledged these aspects as favorable consequences when screening is offered to girls. The importance of extended family screening was illustrated by a study from the Kennedy Krieger Institute, where the screening of 4169 at-risk family members led to the identification of 594 additional males with ALD, of which 250 were presymptomatic [30,31]. Another study from this institute showed that 39 out of 49 boys (80%) identified through extended family screening had unrecognized adrenocortical insufficiency [32]. These aspects of NBS may not result in life-saving treatment for the tested newborn, but they are perceived as important health advantages in a broader sense. Van Dijk et al. investigated the attitudes of different stakeholders, including healthcare providers, test developers and policy makers, towards the expansion of NBS in the Netherlands [16]. The authors identified two perspectives towards NBS: a targeted scope perspective, where only direct health gain to the newborn was prioritized (the newborn being the only beneficiary), and a broader-scope perspective, where secondary effects (such as parents’ reproductive planning) were also considered advantages. According to the authors, in order for stakeholders to feel heard, both perspectives should be taken into account when deciding to screen for specific diseases.

Our study identified a wide range of reasons why patients with ALD preferred to screen only boys or both boys and girls, highlighting the need for further exploration. Respondents in the studies of van der Pal et al. and Schaller et al. described how having more information would enhance their ability to cope with the condition [18,20]. In the open-text responses of the current work, many respondents in favor of screening both boys and girls also emphasized the benefits of early knowledge about their condition before symptoms appear. Respondents in favor of screening both boys and girls agreed with all statements related to screening girls for ALD. Respondents in favor of only screening boys disagreed with some of these statements but responded in a neutral or slightly positive manner to others. This implies that respondents in favor of only screening boys do agree with some of the advantages of also screening girls. However, these potential advantages appear to be insufficient for them to support screening both sexes. The disparities within this group highlight the delicate balance between the public health/medico-ethical perspective on the one hand and the ALD experiential perspective on the other hand (as shown in Table 1). Interestingly, no financial arguments against ALD screening were provided in the open-text responses of our study. It is plausible that even individuals with ALD are not well aware of the impact that an early diagnosis of a genetic disease can have on eligibility for life insurance, the costs for society and other detrimental financial consequences.

It remains uncertain whether background characteristics influence screening preference and, if so, which. The variables age, death of a family member due to ALD, perceived disease severity and having an alternative lifestyle did not exhibit significant differences between the groups. Prior to our study, we hypothesized that women would be more likely to support screening both boys and girls, as female patients may have a better understanding of the potential consequences of ALD on their own health. Female patients were slightly less inclined to support screening only boys compared to males (18% vs. 21%). However, due to the lack of a statistically significant differences in screening preference between sexes, this could be the result of our sample size.

This study is one of the first to evaluate the attitudes of patients towards the addition of ALD to NBS. The Dutch ALD cohort is a large group of men and women with experiential knowledge. We obtained a relatively high response rate of 61% from this group of important stakeholders. The questionnaire we used was constructed after careful consideration and advice by experts in the field of ALD and screening studies. We, therefore, feel that it is a valid instrument for the purpose of this study. The PANDA study evaluated the attitudes of a general population of parents of newborns towards the addition of multiple disease scenarios to NBS, including ALD [18]. Because we used the same questions to evaluate screening preferences, we were able to compare results between studies. This allowed us to draw the important conclusion that the attitudes towards NBS for ALD of a general population appear to be similar to those of ALD patients.

Some limitations pertain to this study as well. Despite achieving a high response rate from patients in our cohort, it is possible that the participants may not be fully representative. Notably, a relatively large proportion of participants had higher levels of education, which may have biased the results. Respondents who did not return the questionnaire may have had less strong screening preferences, which may have skewed levels of agreement. Furthermore, while the Dutch ALD cohort is the largest in the Netherlands and encompasses a large part of the patient population, it may not fully represent the entire population. Ideally, a larger number of patients, particularly in the smaller female subgroup, should have been included to reinforce our findings. The reasons described in this paper on why respondents preferred a type of screening provide valuable insights. The interpretation of the answers may, however, have been somewhat biased. For example, when a respondent who indicated that they prefer screening both boys and girls indicates that the most important reason is early treatment, this answer could refer to the treatment of boys, girls or both sexes. Qualitative interviews with patients could provide more in-depth information on these responses. Moreover, the extent to which these findings can be generalized to patients beyond the Netherlands remains somewhat uncertain. Replicating this survey study in different countries and comparing the results would be insightful.

In conclusion, this study highlights the perspective of individuals with ALD towards (sex-specific) NBS for ALD, with most respondents supporting screening both boys and girls. While we do not aim to provide a recommendation regarding who should be screened for ALD, the results of this study can inform stakeholders involved in the implementation of newborn screening programs. ALD patients are important stakeholders who can provide valuable contributions and insights to the decision-making process. It is important for healthcare professionals involved in newborn screening policy decisions to be aware that sex-specific screening and the screening of both boys and girls can have advantages and disadvantages from the patients’ perspective. Furthermore, it is noteworthy that males and females share similar considerations on this matter. These conclusions hold particular relevance as an increasing number of countries consider adding ALD to their newborn screening programs.

## Figures and Tables

**Table 1 IJNS-09-00051-t001:** Overview of different aspects related to the public health/medico-ethical perspective and the ALD experiential perspective with regard to sex-specific newborn screening for ALD.

Public Health/Medico-Ethical Perspective	ALD Experiential Perspective
No treatment for myelopathy (early detection of ALD in girls therefore not eligible)	Early detection of ALD in girls can increase early disease awareness and anticipate future health problems
Late onset of myelopathy can make identified asymptomatic girls ‘patients in waiting’	Early detection of ALD in girls shortens the time to diagnosis when symptoms arise (reducing ‘diagnostic odyssey’)
Right not to know (right to an open future)	Early detection of ALD in girls can enable parents to make informed decisions regarding family planning
Negative impact on psychological functioning and quality of life of early detection of ALD being an untreatable late-onset disorder in girls	Early detection of ALD in girls allows them to make informed decisions about the risk of having children with ALD at an adult age
Possible financial consequences of receiving a genetic diagnosis (e.g., eligibility for life insurance)	Early detection of ALD in girls enables extended family screening
Loss of the happy/“golden” years when parents are informed of an untreatable condition in their child	

ALD: adrenoleukodystrophy.

**Table 2 IJNS-09-00051-t002:** Background of the respondents.

	All Patients (n = 66)	Male Patients (n = 38)	Female Patients (n = 28)
Age in years, mean + SD (range)	49 ± 16 (20–76)	46 ± 17 (20–76)	54 ± 14 (25–76)
Patient age of diagnosis, n (%)			
Pediatric age ^a^	18 (27%)	14 (37%)	4 (14%)
Adult age	43 (65%)	24 (63%)	19 (68%)
Unknown	1 (2%)	0 (0%)	1 (4%)
Missing	4 (6%)	0 (0%)	4 (14%)
Patient route of diagnosis, n (%)			
ALD related symptoms	20 (30%)	18 (47%)	2 (7%)
Family member tested positive	42 (64%)	20 (53%)	22 (79%)
Missing	4 (6%)	0 (0%)	4 (14%)
Reported symptoms, n (%)			
(Arrested) cerebral ALD	2 (3%)	2 (5%)	0 (0%)
Adrenal insufficiency	18 (27%)	18 (47%)	0 (0%)
Myelopathy	36 (55%)	22 (58%)	14 (50%)
Other	4 (6%)	1 (3%)	3 (11%)
Asymptomatic	18 (27%)	6 (16%)	12 (43%)
I do not know	2 (3%)	1 (3%)	1 (4%)
Missing	4 (6%)	0 (0%)	4 (14%)
Reported family members with ALD, n (%)			
Son(s)	12 (18%)	0 (0%)	12 (43%)
Daughter(s)	11 (17%)	7 (18%)	4 (14%)
Father	6 (9%)	0 (0%)	6 (21%)
Mother	44 (67%)	32 (84%)	12 (43%)
Brother(s)	26 (39%)	18 (47%)	8 (29%)
Sister(s)	18 (27%)	12 (32%)	6 (21%)
Grandson(s)/granddaughter(s)	2 (3%)	1 (3%)	1 (4%)
Other	17 (26%)	10 (26%)	7 (25%)
None	1 (2%)	1 (3%)	0 (0%)
Deceased family members due to ALD, n (%)			
Yes	34 (52%)	20 (71%)	14 (37%)
No	31 (47%)	8 (29%)	23 (60%)
Missing	1 (2%)	0 (0%)	1 (3%)
Education level ^b^, n (%)			
Low	16 (24%)	7 (18%)	9 (32%)
Middle	16 (24%)	8 (21%)	8 (29%)
High	30 (46%)	20 (53%)	10 (36%)
Missing	4 (6%)	3 (8%)	1 (4%)
Religion, n (%)			
Unreligious	28 (42%)	18 (48%)	10 (36%)
Not active within religion	25 (38%)	13 (34%)	12 (43%)
Somewhat active within religion	5 (8%)	2 (5%)	3 (10%)
Active within religion	4 (6%)	2 (5%)	2 (7%)
Missing	4 (6%)	3 (8%)	1 (4%)
Alternative lifestyle ^c^			
Yes	17 (25%)	7 (18%)	10 (36%)
No	47 (70%)	30 (79%)	17 (61%)
Missing	4 (5%)	1 (3%)	1 (3%)
Vaccination status of children			
All children fully vaccinated	47 (71%)	26 (68%)	21 (75%)
Children partially vaccinated	7 (11%)	3 (8%)	4 (14%)
Children not vaccinated	0 (0%)	0 (0%)	0 (0%)
No children/no child wish	6 (9%)	4 (11%)	2 (7%)
Missing	6 (9%)	5 (13%)	1 (4%)

ALD: adrenoleukodystrophy; ^a^ Pediatric age ≤ 18 years old; ^b^ Low education level = Elementary school, lower level of secondary school, and lower vocational training. Middle education level = higher level of secondary school and intermediate vocational training. High education level = High vocational training and university. ^c^ Alternative lifestyle was surveyed by asking whether respondents valued homeopathy, anthroposophy, alternative medicine or another alternative lifestyle.

**Table 3 IJNS-09-00051-t003:** Attitudes of respondents towards sex-specific newborn screening for ALD.

	All Respondents (n = 66)	Male Respondents (n = 38)	Female Respondents (n = 28)
In favor of only screening boys for ALD, n (%)	13 (20%)	8 (21%)	5 (18%)
In favor of screening boys and girls for ALD, n (%)	53 (80%)	30 (79%)	23 (82%)

ALD: adrenoleukodystrophy.

**Table 4 IJNS-09-00051-t004:** Agreement with statements related to screening preference.

Statement	All Patients n = 66	In Favor of ALD Screening Only Boys (in Favor of Sex-Specific Screening) n = 38	In Favor of ALD Screening Boys and Girls (Opposed to Sex-Specific Screening) n = 28
**Arguments in favor of ALD screening newborn *boys***			
ALD should be detected early in boys so that they can receive optimal treatment immediately upon the onset of the first symptoms.	4.5 (±0.9)	4.5 (±0.5)	4.5 (±1.0)
ALD should be detected early in boys because it can prevent a long period between the onset of the first symptoms and the eventual diagnosis.	4.4 (±1.1)	4.2 (±1.1)	4.4 (±1.0)
ALD should be detected early in boys because it enables the identification of other family members with ALD.	4.3 (±0.9)	4.1 (±0.5)	4.2 (±1.0)
ALD should be detected early in boys so that parents can be informed in time about possibilities for further family planning.	4.1 (±1.2)	4.2 (±0.9)	4.1 (±1.3)
**Arguments in favor of ALD screening newborn *girls***			
ALD should be detected early in girls because it enables the identification of other family members with ALD (including boys and men).	4.1 (±1.0)	3.4 (±0.9)	4.3 (±0.9)
ALD should be detected early in girls so that parents can be informed in time about possibilities for further family planning.	4.0 (±1.2)	3.5 (±1.0)	4.1 (±1.3)
ALD should be detected early in girls so that they can receive optimal treatment immediately upon the onset of the first symptoms.	3.6 (±1.2)	3.0 (±0.8)	3.8 (±1.2)
ALD should be detected early in girls because it can prevent a long period between the first symptoms and the eventual diagnosis.	3.6 (±1.2)	2.9 (±1.0)	3.8 (±1.2)
It is unfair for girls not to be tested for ALD, while boys are.	3.6 (±1.3)	2.3 (±0.9)	3.9 (±1.1)
**Arguments opposed to ALD screening newborn *girls***			
ALD should NOT be detected early in girls because it can be mentally burdensome to know that they may develop untreatable symptoms later in life.	2.3 (±1.2)	3.6 (±0.8)	1.9 (±1.0)
ALD should NOT be detected early in girls as it adds little to their quality of life.	2.2 (±1.2)	3.6 (±1.0)	1.8 (±0.9)
ALD should NOT be detected early in girls because there is no treatment available.	2.2 (±1.2)	3.8 (±1.0)	1.8 (±0.9)
ALD should NOT be detected early in girls, because they almost never develop life-threatening symptoms.	2.1 (±1.2)	3.5 (±1.1)	1.7 (±0.9)
**Arguments opposed to ALD screening newborns *in general***			
ALD should NOT be detected early as the diagnosis can have adverse financial consequences, such as when applying for insurance.	2.3 (±1.1)	2.8 (±0.9)	2.1 (±1.1)
ALD should NOT be detected early because every child has the right to an ‘open’ future (and therefore the right not to know that he/she has ALD).	1.8 (±0.8)	1.8 (±0.9)	1.7 (±0.8)
ALD should NOT be detected early as it deprives parents of the opportunity to enjoy a (still) healthy baby.	1.8 (±0.9)	1.8 (±0.6)	1.8 (±1.0)
ALD should NOT be detected early because you have to take life as it comes.	1.5 (±0.8)	1.6 (±0.5)	1.5 (±0.8)
ALD should NOT be detected early as it impairs bonding between parents and their child.	1.4 (±0.7)	1.5 (±0.5)	1.4 (±0.8)
Early detection of ALD through the heel prick test in both boys and girls is too expensive for society.	1.4 (±0.7)	1.5 (±0.5)	1.4 (±0.7)

Agreement to statements on a 5-point scale are presented as mean (±standard deviation). Higher means indicate higher agreement. Results are organized from highest to lowest mean level of agreement for all patients within that category of arguments.

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
