# Peer review of "Attitudes of Patients with Adrenoleukodystrophy towards Sex-Specific Newborn Screening"

_2409-515X, 2023, doi:10.3390/ijns9030051_

Round 1
Reviewer 1 Report
Thank you for the opportunity to review this manuscript, which describes the opinions for towards gender-specific newborn screening for adrenoleukodystrophy. Firstly, I congratulate the authors on this initiative as it provides some direction for newborn screening jurisdictions in their implementation of X-ALD screening.
The reason for this questionnaire, is not about whether or not to screen for X-ALD, but whether or not to screen females for X-ALD. The pertinent information, in my opinion, is from females (and not males) and the manuscript should be updated throughout to the female responses.
There are only n=28 females who have responded, with 18% saying no to screening, and I am not sure that there is sufficient power in these results to provide guidance for screening policy.
As the number of respondents is small it would be beneficial to circulate this questionnaire more broadly to gain more female responses. In addition, it could be worthwhile having a control cohort of non affected females to gauge society opinion. As a minimum, I would refocus the manuscript on female responses.
Other minor points:
- There are a number of minor grammatical errors throughout that need addressing in the revised manuscript.
- Significant figures in table 2 - adjust to max one decimal place (ie based on the SD’s provided)
- There are a number of minor grammatical errors throughout that need addressing in the revised manuscript.
Author Response
Thank you for the opportunity to review this manuscript, which describes the opinions for towards gender-specific newborn screening for adrenoleukodystrophy. Firstly, I congratulate the authors on this initiative as it provides some direction for newborn screening jurisdictions in their implementation of X-ALD screening.
Response: We thank the reviewer for these kind words. Indeed, we hope that our manuscript will provide direction in the debate on sex-specific screening for ALD.
The reason for this questionnaire, is not about whether or not to screen for X-ALD, but whether or not to screen females for X-ALD. The pertinent information, in my opinion, is from females (and not males) and the manuscript should be updated throughout to the female responses.
Response: we thank the reviewer for this comment. Although we agree that the responses from females are of vital importance, their opinion does not seem to differ much from males. This is demonstrated by the non-significant difference between screening preferences of men and women. We performed independent t-tests with sex as the independent variable for all the statements to explore differences in motives between men and women.
We modified the manuscript and have added the following text (Materials and Methods, line 128-130): “In order to explore differences in levels of agreement between sexes, we performed in-dependent t-tests for each statement”; (Results, line 187-188): “Independent t-tests for each statement revealed no significant differences in the mean levels of agreement between male and female respondents (data not shown).”, and (Discussion, lines 322-323) “Furthermore, it is noteworthy that males and females share similar considerations on this matter.”
There are only n=28 females who have responded, with 18% saying no to screening, and I am not sure that there is sufficient power in these results to provide guidance for screening policy.
Response: We agree that this group is too small to draw conclusions on the consensus of the entire female ALD population. With this manuscript, we do not aim to draw a definite conclusion on who should be screened for ALD. Rather, we hope to provide insights on the preferences and motives of these important stakeholders to continue the debate on sex-specific screening. We have added a sentence to the Discussion in the section “some limitations” (lines 304-305) “Ideally, a larger number of patients, particularly in the smaller female subgroup, would have been included to reinforce our findings.”
As the number of respondents is small it would be beneficial to circulate this questionnaire more broadly to gain more female responses. In addition, it could be worthwhile having a control cohort of non affected females to gauge society opinion. As a minimum, I would refocus the manuscript on female responses.
Response: We agree that a larger number of participants would have helped to affirm our conclusions. Sadly, as ALD is a rare disease, inclusion of a large number of participants is difficult. In the future, we plan to expand this survey with the aim to reach a higher number of female respondents. However, we feel that the responses from a general group of parents (n=804) of healthy newborns (as was described in the PANDA study by van der Pal et al., 2022) provides some level of insights on the views of a general population.
We have added the following text to the Discussion (lines 294-296): “This [the results by van der Pal et al] allowed us to draw the important conclusion that the attitudes towards NBS for ALD of a general population appear to be similar to those of ALD patients.”
Other minor points:
- There are a number of minor grammatical errors throughout that need addressing in the revised manuscript.
Response: English grammar was revised throughout the manuscript and corrected accordingly.
- Significant figures in table 2 - adjust to max one decimal place (ie based on the SD’s provided)
Response: We appreciate the reviewer's suggestion to limit the significant figures in Table 2 to a maximum of one decimal place, consistent with the provided standard deviations. However, upon reviewing the original Table 2, we noted that no decimal points are present in the table.
Reviewer 2 Report
This is a very interesting study on the perspectives of X-ALD patients towards NBS for X-ALD, especially towards question of sex-selective NBS. It provides better understanding of patient perspectives on this complex issue. The paper is nicely written.
Several minor suggestions for authors:
- Table 1 - How is "Experiential" defined? There is another (partly overlaping with "Public Health") perspective which could be called "(Medico-)Ethical" perspective - could you possibly add another column summarizing specific medico-ethical aspects on this particular issue. E.g. Screening of adult-onset disorders in children is ethically problematic etc.
- Possibly some additional literature on medico-ethical aspects of NBS for adult-onset disorders could be included and briefly summarized (e.g. in Table 1 and Discussion).
- Methods - Questionnaire could be added as Supplementary file.
- Results: Table 2 might be removed to the Supplementary files, since it contains many figures that are not of direct importance to the study and are already briefly summarized in the text.
- Discussion: This section might be shortened (condensed) a bit if possible, some "balast" sentences are present. These results might be hard to generalize to all countries, this notion should be added to Conclusions - actual international comparisons might be beneficial.
Author Response
This is a very interesting study on the perspectives of X-ALD patients towards NBS for X-ALD, especially towards question of sex-selective NBS. It provides better understanding of patient perspectives on this complex issue. The paper is nicely written.
Response: We thank the reviewer for the appreciation of our manuscript. We hope that it further clarifies the important patient perspective.
Several minor suggestions for authors:
- Table 1 - How is "Experiential" defined?
Response: We added the following text (line 58-59): (…) the “ALD experiential perspective”, which is shaped by personal experiences, participation in patient organizations or by the experiences of relatives, ….
- There is another (partly overlaping with "Public Health") perspective which could be called "(Medico-)Ethical" perspective - could you possibly add another column summarizing specific medico-ethical aspects on this particular issue. E.g. Screening of adult-onset disorders in children is ethically problematic etc. Possibly some additional literature on medico-ethical aspects of NBS for adult-onset disorders could be included and briefly summarized (e.g. in Table 1 and Discussion).
Response: We thank the reviewer for this suggestion and introducing the medico-ethical perspective. As the reviewer already states, this perspective has significant overlap with the public health perspective. The choice to include a disease in newborn screening is based on medico-ethical considerations and we feel that these considerations (such as non-maleficence, patient autonomy and justice) are all integral parts of the Wilson and Jungner criteria. We have therefore added the term medico-ethical considerations to the text (lines 59-60) “….and the public health/medico-ethical perspective guided by the Wilson and Jungner criteria” as well as into the legend and heading of Table 1 “Public Health/Medico-ethical Perspective”, and the Discussion (line 270).
- Methods - Questionnaire could be added as Supplementary file.
Response: We have added the original questionnaire as a supplementary file.
- Results: Table 2 might be removed to the Supplementary files, since it contains many figures that are not of direct importance to the study and are already briefly summarized in the text.
Response: We appreciate the suggestion to consider moving Table 2 to the Supplementary files due to its extensive content. However, we've observed that supplemental data often doesn't receive as much attention as the main paper content (in general only the main paper is downloaded and stored by readers). If it is no problem, we would like to keep the Table in the main paper.
- Discussion: This section might be shortened (condensed) a bit if possible, some "balast" sentences are present. These results might be hard to generalize to all countries, this notion should be added to
Conclusions - actual international comparisons might be beneficial.
Response: We have revised the discussion and removed some redundant sentences.
We have also added the following text (lines 311-313): “Moreover, the extent to which these findings can be generalized to patients beyond the Netherlands remains somewhat uncertain. Replicating this survey study in different countries and comparing the results would be insightful.”Round 2
Reviewer 1 Report
Thank you for the opportunity to re-review this X-ALD questionnaire manuscript. The authors have addressed all points raised except for the point about Table 2 and the number of decimal places. However, I see that I should have listed Table 4 on this point and apologize for this typographical error on my part. Can you please review Table 4 - Significant figures, - adjust to max one decimal place (ie based on the SD’s provided)
Author Response
We thank the reviewer for pointing out it's Table 4. We have corrected the figures to 1 decimal point